# Blockchain-Based Deep CNN for Brain Tumor Prediction Using MRI Scans

**DOI:** 10.3390/diagnostics13071229

**Published:** 2023-03-24

**Authors:** Farah Mohammad, Saad Al Ahmadi, Jalal Al Muhtadi

**Affiliations:** 1Center of Excellence in Information Assurance (CoEIA), King Saud University, Riyadh 11543, Saudi Arabia; 2College of Computer & Information Sciences, King Saud University, Riyadh 11543, Saudi Arabia

**Keywords:** brain tumor, deep learning, blockchain, secure CNN

## Abstract

Brain tumors are nonlinear and present with variations in their size, form, and textural variation; this might make it difficult to diagnose them and perform surgical excision using magnetic resonance imaging (MRI) scans. The procedures that are currently available are conducted by radiologists, brain surgeons, and clinical specialists. Studying brain MRIs is laborious, error-prone, and time-consuming, but they nonetheless show high positional accuracy in the case of brain cells. The proposed convolutional neural network model, an existing blockchain-based method, is used to secure the network for the precise prediction of brain tumors, such as pituitary tumors, meningioma tumors, and glioma tumors. MRI scans of the brain are first put into pre-trained deep models after being normalized in a fixed dimension. These structures are altered at each layer, increasing their security and safety. To guard against potential layer deletions, modification attacks, and tempering, each layer has an additional block that stores specific information. Multiple blocks are used to store information, including blocks related to each layer, cloud ledger blocks kept in cloud storage, and ledger blocks connected to the network. Later, the features are retrieved, merged, and optimized utilizing a Genetic Algorithm and have attained a competitive performance compared with the state-of-the-art (SOTA) methods using different ML classifiers.

## 1. Introduction

Brain tumor disorders, which seriously impair people and pose a threat to their lives, have recently appeared to gain more attention. Brain cancer is the tenth most common cause of death in both men and women [1]. According to the International Agency for Research on Cancer, around 97,000 individuals worldwide die from brain tumors each year, and approximately 126,000 people are diagnosed with them [2]. On the other hand, the chances of surviving malignant brain tumors vary considerably and are influenced by several variables, including the patient’s age and the kind of tumor. White matter (WM), grey matter (GM), and cerebral spinal fluid (CSF) are the three basic tissue types that make up the average brain tissue, in contrast to aberrant tissues, such as tumors, necrosis, and edema.

Edema develops close to the margins of an active tumor, while necrosis occurs within the tumor itself [3]. While benign tumors develop slowly and never invade or spread to other tissues, malignant destructive tumors grow swiftly [4]. Based on these two major classifications, brain tumors are divided into three types: gliomas, meningiomas, and pituitary tumors. Glioma tumors develop in several parts of the brain tissues, rather than blood vessels and nerve cells. Pituitary tumors develop inside the skull, and meningioma tumors develop on the membrane area that surrounds and impacts the central nervous system and brain [5]. The World Health Organization has recognized many types of brain tumors. Based on the cell’s origin and behavior, which might range between being less aggressive and more aggressive [6], this classification is made. Meningiomas in benign form are slow-growing tumors, whereas malignant forms are gliomas, in contrast to pituitary tumors, which, even if minor, can still result in various health issues [7]. These three tumor types differ most significantly from one another. Considering the facts presented above, identifying these three different tumor types is an important step in the clinical diagnosis process for patients.

For many years, medical professionals have asserted that it is challenging to use clinical imaging and human interpretation to detect brain cancers. More trustworthy early tumor detection methods, including computer-aided diagnosis (CAD) [8], are therefore urgently needed. Applications in medical diagnosis that depend on feature extraction from medical images, such as attempting to differentiate between healthy and pathological tissue, have relied heavily on CAD techniques [9,10,11]. Techniques for medical imaging are essential for early tumor detection and for improving treatment options. Brain tumors are studied using a variety of non-invasive imaging techniques, such as CT, PET, MRI, SPECT, and X-ray [12,13]. A crucial stage of treatment is the employment of non-invasive technologies to find brain tumors [14]. To help in diagnosis, medical imaging techniques can reveal details on the location, size, shape, and classification of brain tumors [15]. MRI is recognized as a standard instrument for providing specific information on the morphological tissue of people due to its extensive capacity to capture the precision of soft tissue in comparison to other medical image modalities [16].

The proper diagnosis of brain tumors can be complicated and challenging due to several potential interfering factors. The industry-recognized imaging method for use in the diagnosis of brain tumors is magnetic resonance imaging (MRI) [17]. Image anomalies, however, can result from a variety of factors, such as patient motion, metallic implants, and other technological issues. The size and location of a brain tumor may also influence the accuracy of the diagnosis. Certain tumors may be difficult to detect on imaging tests that are challenging to access or operate on. Moreover, tiny tumors could be more difficult to locate, while larger tumors might be mistaken for other conditions, including infections or strokes [18]. Pathologists’ interpretations of the outcomes of biopsies can be arbitrary and differences can occur between them. There may be a symptom overlap between brain tumors and other health issues, including headaches, infections, or strokes. Age, impairments, and prescription use are all patient-related factors that may affect the diagnosis [19].

Deep learning (DL), a machine learning (ML) approach, has gained prominence and a great deal of interest in each domain, particularly in medical image analysis [20]. Deep learning can analyze a lot of imbalanced data by passing it through numerous layers; each layer can extract features incrementally and transfer them to the next layer, giving deep learning more power and flexibility [21,22]. The focus on automatically deriving features that represent data interpretations is the deep learning technique’s defining feature. Convolutional neural networks (CNN) are the most frequently used deep learning techniques in the field of medical image processing. Utilizing autonomously extracted structured data representations and features from medical images, the CNN model provides an effective processing capacity [23,24,25].

A Secure CNN model, known as Secure CNN, was created using blockchain technology in CNNs. At each phase, CNN’s design is changed to increase the security and safety. We used an already-in-use strategy for this [26]. Every CNN model layer has an additional block that keeps certain data to prevent layer deletion, modification assaults, and tampering. These data consist of the following: (a) inputs and outputs of the preceding and current layers that are encrypted; (b) public keys for all layers and the private keys for the layers around them; and (c) the weights of the current and following layers. These data are kept in several blocks, including a block connected to each layer, a block connected to the network, and a cloud ledger block kept connected to the cloud storage. The suggested CNNs can detect any tampering, whether it be at the parameter or network level, and take preventative measures to avoid it. Later, we additionally use the Genetic Algorithm (GA) to refine the CNN models’ features, and the refined features are passed to ML classifiers. The major contribution of the proposed work is as follows:Blockchain layers have been added to the CNN models to secure the input and output.Blockchain-based secure CNN models have been fine tuned for feature engineering.The derived features are fused and optimized using a finetuned genetic algorithm.

The rest of the manuscript is organized as follows: In Section 2, a quick overview of the contemporary deep learning techniques for predicting brain tumors are covered. Section 3 provides a comprehensive description of the suggested technique. The results from the simulations and experiments are shown in Section 4. Section 5 presents the discussion, followed by the conclusion is Section 6.

## 2. Related Work

There has been a significant amount of work conducted on medical image analysis, and numerous researchers have made contributions to different sub-fields of medical imaging. This section examines the previous research on the classification and detection of brain tumors. Most of the current research in medical imaging focuses on the automatic segmentation and classification of the tumor region in MR images. Researchers [27] have modified the AlexNet model [28] to categorize brain MR images into healthy and unhealthy. The unhealthy images were further classified into low-grade and high-grade images. The technique showed promising results, with 91.16% classification accuracy. Özyurt et al. [29] introduced an approach for the classification of MR images. First, they segment the tumor area from the images as malignant and benign using the expert maximum fuzzy-sure entropy method. Then, the features from segmented images are extracted using a CNN model and classified using SVM and KNN classifiers. The classification results were obtained using 5-fold cross-validation.

The authors of [30] presented a technique for classifying brain tumors into different grades using deep CNN and extensive data augmentation techniques. They trained a deep CNN on the augmented dataset and evaluated its performance on a test set of MRI scans. The results showed that CNN was able to classify brain tumors into different grades with high accuracy and that the use of data augmentation techniques improved the performance of the model. In [31], the authors used five pre-trained models (Xception, ResNet50, InceptionV3, VGG16, and MobileNet) to train their brain tumor dataset, and evaluated the performance of each model using the F1-score measure on unseen images. The results showed that all five models achieved high accuracy, with the highest score of 98.75%. The authors concluded that these high accuracy rates have the potential to improve the early detection of tumors and prevent disability caused by advanced tumors. The proposed work [32] focuses on using computer-assisted techniques to improve the deciphering of medical imaging, specifically using brain MRI images for the identification of tumors. The authors proposed the use of deep learning techniques to classify brain tumors. They test various CNN architectures, including basic CNN and VGG-16, and report that their designed model has an accuracy of 95.71% on an online dataset.

Rinesh et al. [33] proposed a technique that combines k-Mean clustering-based processes to locate the tumor; techniques such as the k-nearest neighbor and k-means clustering may be used, with the value of k being calculated by an optimization technique known as the firefly method. They additionally name the brain regions using a multilayer feedforward neural network. The proposed method, which has a lower mean absolute error value and a higher peak signal-to-noise ratio than the existing methods, such as parallel k-means clustering and hybrid k-means clustering, was found to generate better results. Overall, 96.47% accuracy, 98.24% specificity, and 96.32% sensitivity were attained by the suggested model. The researchers in [34] utilized transfer learning to extract the characteristics from a convolutional neural network that has been built for deep brain magnetic resonance imaging scans. To assess the performance, multiple layers of separate CNNs are created. The created CNN models are then utilized to train several MLCs by transferring deep features. Compared to the existing, widely used pre-trained deep-feature MLC training models, the suggested CNN deep-feature-trained support vector machine model produced better accuracy, with 98% accuracy in detecting and distinguishing brain tumors.

The study [35] developed an effective approach using the VGG16 model to assist in making quick, effective, and precise judgments by employing MRI to find brain cancers. The method was tested on a dataset of 253 MRI brain pictures used to diagnose brain tumors, 155 of which had a tumor. The proposed work in [36] aimed to improve the classification process for detecting brain tumors using machine learning algorithms. Six different machine learning algorithms were used: Random Forest (RF), Naive Bayes (NB), Neural Networks (NN), CN2 Rule Induction (CN2), Support Vector Machine (SVM), and Decision Tree (Tree). The results were collected using different performance measures, such as classification accuracy, the area under the Receiver Operating Characteristic (ROC) curve, precision, recall, and F1 Score (F1). A 10-fold cross-validation technique was used to strengthen the training and testing process. The results show that the maximum classification accuracy of 95.3% was achieved on SVM. Habiba et al. [37] presented a study on the detection and classification of brain tumors using deep learning-based classifiers that extract features from MRI images. The researchers used a publicly available dataset and a transfer learning approach with the InceptionV3 and DenseNet201 models. Data augmentation was applied to improve the classification results and avoid overfitting. The proposed “Brain-Deep Net” model, a deep convolutional neural network, consisted of six densely connected convolution layers, which extracted features from dense layers. The dense layers can extract features more efficiently from brain MRI than the other models. The model successfully distinguished between the three most frequent forms of brain tumors—glioma, meningioma, and pituitary—with a classification accuracy of 96.3%.

Several ML- and DL-based approaches have been presented in the literature for the accurate prediction of brain tumors from MRI scans. The main goal of the presented approach was to adopt the most robust approach with a higher recognition rate by coping with the computational cost. The presented literature covers the most state-of-the-art (SOTA) approaches of DL to increase the diagnosis process of brain tumors.

## 3. Proposed Methodology

### 3.1. Blockchain

A decentralized system, called blockchain (BC), employs distributed ledgers to track various user transactions [38]. These users may be systems, people, or even algorithms. The transactions are saved permanently, unchangeable, and simple to verify upon a single request. The fundamental components of numerous cryptocurrencies have been built using BC technology. There is no clear connection between BCs and Convolutional Neural Networks (CNNs). Nevertheless, several real-time applications, such as machine security, healthcare, and surveillance, these technologies can provide a more secure structure. BC’s strengths are its Transitive Hash, Encryption at every level, and Decentralization.

Any algorithm tempering, such as feature derivation, concatenation, feature mapping, and feature optimization, is forbidden by transitive hashes and encryption schemes. To highlight an illegal change at a particular node or layer of the algorithm, transitive hashes will search for any change at any level. Once found, it is possible to return a node or layer to its initial state. As a result of its decentralized structure, the algorithm cannot be tricked at any level by anyone and is not stored entirely on a single network. A secure and safe CNN might be suggested using these characteristics. Blockchain is therefore a top contender for secure and safe CNN. You can use symmetric or asymmetric key algorithms to encrypt the data. Symmetric encryption techniques have a flaw because they only employ one key for the encryption and decryption of a message. Anyone in possession of the key can quickly decrypt the message and make the necessary changes or deletions. Two keys—one public and one private—are employed in an asymmetric algorithm to encrypt and decrypt plain text [39]. While the public key is widely shared, the private key is kept private. Anyone can use their public key to encrypt a message for the recipient, but only they have access to their private key to decrypt it. Asymmetric encryption increases security even though it slows down the procedure overall [40]. When using CNNs with BC enabled, asymmetric encryption (AE) is used.

A piece of software known as a Smart Contract (SC) guarantees reliable and genuine transactions. SCs are also used to keep track of the start and end of each transaction. These SCs’ main advantage is their lack of external API requirements, which makes it impossible for any other external agent to pollute the data. SCs can be installed on CNNs at many levels to increase security and safety. Using an SC, all the network’s inputs and outputs can be preserved in a ledger, where they can later be verified or restored. The proposed CNN creates several SCs, referred to as Layer Ledger Blocks (LLB) for each layer, which store the data for the current and following layers. With all the network layer data, a new SC called the Central Ledger Block (CLB) is created. The local storage for CLB is kept on the network, and the cloud storage is kept with a copy of CLB as well. The local CLB and the cloud based CLB are continuously synchronized. To prevent an intruder from determining the order of the layers, the LLBs update the CLB at random. The Secure CNN structure is presented in Figure 1.

Blocks are connected in the BC architecture by utilizing an incremental linked list, which served as the model for the Secure CNN’s structure. The sole distinction is that ledgers and blocks can exist indefinitely in BC technology. Furthermore, there are only a certain number of blocks and ledgers in the Secure CNN, and these numbers are solely derived from CNN layers. Each layer of the network has a ledger block that does the following: stores the layer’s parameter information; computes the output of the layer; validates the output of the layer; and updates both the layer’s ledger block and the central ledger block. The structure [41] of the layer ledger block is shown in Figure 2.

The LLB is nothing more than an additional layer of a CNN model with zero bias, an identity function as an activation function, and an identical weight matrix. As a result, the output of LLB will be its input. The LLB includes hashes of the current and previous layers, the current layer’s private and public keys, the next layer’s encrypted layer parameters, and the immediately preceding and next levels’ public keys. Using the renowned AE algorithm Data Encryption Standard (DES), hash creation and parameter encryption are accomplished [42]. The overall organization [41] of the central ledger block is shown in Figure 3.

The entire method is handled as one transaction once one layer has been executed and the outcomes are fed to the next layer. Each transaction is associated with a signature in the CLB, which holds the data on all transactions at random. These data have been randomized to increase their security from tempering. The shared storage that also holds a model’s state is the central ledger block. Using the parameters for the current and prior layers, as well as the preceding layer’s hash, a hash at a particular layer is calculated. The hash (h) of the layer, if it is the current layer, can be determined as follows:(1)$$h_{i}=\rho \left(h_{i-1},parametrs_{i}-parametrs_{i-1}\right),$$

In the above Equation (1), ρ illustrates the data encryption standard (DES) method. When tampering occurs, the hash keys are kept in a central ledger block to determine which layer has been tempered. Every layer’s information is randomly saved in the central ledger block. Even the layers in the core block are unaware of their sequential ordering. The authenticity of parameter in the LLB is validated from the output and input of the corresponding hash key layer. True or false are the two possible values for this authenticity attribute. The layer will pass the output to the following layer if the value is set to true. Unless the result is false, the network has been compromised, and the output will no longer be sent to the following layer. It recalculates the hashes and returns the previous and current layers’ parameters. Unless validity becomes legitimate once more, this process will continue.

Following the authenticity verification, the CLB executes the following operations: (a) DES encryption of the layer output using the public key of the subsequent layer; (b) attachment of a signature; and (c) computation of the subsequent layer’s hash. Each layer checks if the update is signed-verified or not after each update in the central block. This verification is carried by the layer just underneath. By utilizing the private key from the current layer to encrypt the parameters from the previous layer, it is possible to determine the signature $Sign_{i}$ of any layer i. If the weights are denoted as $X_{i}$, $B_{i}$ is bias, the input of the layer is $I_{i}$, then $P_{i}$ is computed using the activation ρ as $P_{i}=\rho \left(X_{i}\times Y_{i}\right)+B_{i}$. Its output becomes the input of the following layer when it is encrypted with the public key of that layer. $Y_{i+1}=\rho (P_{i}$, $Pub_{i+1})$. The current layer’s hash is calculated as follows, utilizing the parameters of the input layers and the previous layer’s hash: $H_{i}=\sigma \left(H_{i-1},Y_{i},\;Y_{i-1}\right).$ Consider that the present layer modifies the central ledger block by $H_{i},\;Y_{i}$, $P_{i}$, and $Sign_{i}$, The verification process will then be carried out at layer i+1 by utilizing the public key of the preceding layer to decrypt the identity of the preceding layer. The outputs are valid if the signature matches; otherwise, the network has been compromised. That the layer receives the input from the authorized layers is ensured by this signature verification. Moreover, any layer can be validated at any time. For example, say layer i is altered.

The input of the corresponding layer will be considered fabricated if $Y_{i}$ is not equal to the output of the previous layer $P_{i-1}$, or If the public key of the preceding layer’s prior layer cannot be used to decrypt the validity of the previous layer to establish the authenticity of the current layer. This leads us to the conclusion that either the $P_{i}$ is not authentic, inferring that the layer i before it is tempered, or the $Y_{i}$ are not authentic, inferring that the layer i before it is tempered. As the present layer i is tempered, $P_{i-1}$ must be real, because if it had been tempered, the layer would never have been able to produce an output.

### 3.2. Deep Learning Architecture Using Secure CNN

Brain tumors are categorized into relevant classes using the Deep Learning (DL) architecture. The pre-trained models, InceptionV3, GoogleNet, and DenseNet201, make up the DL architecture. According to the suggested technique to extract the image features, these models are converted into Secure CNN models. The mode value-based serial technique is utilized to fuse the extracted features, and GA is employed for the optimization. The best features are chosen through GA and fed to the ML classifiers for the final classification. Figure 4 depicts the whole DL architectural flow.

#### Convolutional Neural Networks

The CNNs were suggested by [43] to categorize the handwritten numbers [44]. The organic organization of the human mind, where neurons transmit information from one cell to another, serves as an inspiration for the CNN models. The speed and precision of a neuron’s activity determine a person’s intellectual potential. Similarly, the success of CNNs depends on its learning and lowering the error rate. The CNN architectures are composed of several layers that conduct various functions at various levels. The CNN models have an input layer that only takes images of a particular size. Multiple stacks of convolutional, pooling, ReLu, and normalization layers are then applied to the input image.

The final layers of the CNN, which mostly consist of the Softmax layer and fully connected layers, are utilized to extract the learned features. Every time we discuss CNNs, the pre-trained networks come up. In contrast to other Machine Learning (ML) networks, these pre-trained networks use pre-processed photos as the input rather than feature vectors. Large datasets, such as ImageNet, are used to train these models in a supervised setting.

*Inceptionv3*: The Large-Scale Visual Recognition Challenge (ILSVRC-2014) was proposed as a better version of it [45]. To increase the adaptability of computer-related applications, the system was designed to reduce computing costs while improving the characterization precision. On the ILSVRC-2012 characterizations, it achieves a best 1 error ratio of 22.0% and a best 5 error ratio of 6.1% [46]. The input for this network, which has 346 layers, is an RGB image with dimensions 299 by 299 by 3. A feature matrix with a dimension of $FV_{1}\times 2048$ was returned by the “avg pool” layer.

*GoogleNet*: In the ILSVRC 2014 [47] classification challenge, the Google Neural Network [45] architecture did well on the classification challenges, taking first place, with a top 5 error of 6.67%, without any training on outside data. The GoogleNet architecture comprises 22 layers (27 layers when pooling layers are included), and there is a total of 9 inception modules inside these layers. The size of the input image to the GoogleNet is 224 × 224 × 3. The feature’s engineered matrix has a size of $FV_{2}\times 1000$.

*DenseNet*: The deep CNN model comprises four dense blocks [48] and each of the first three dense blocks receives a transition layer, while the final dense block receives a classification layer. The first convolutional layer output size is 112 × 112, with stride and filter sizes of 7 × 7 and 2 × 2, respectively. The convolutional layer is followed by a max-pooling layer with a 3 × 3 pooling block having stride 2. The classification layer now includes the fully connected (FC) layer and the global average pool layer with a filter size of 7 × 7. A $FV_{3}$ × 1000-dimension feature matrix was returned by the FC layer. This network accepts an RGB image as the input with a size of 224 × 224 × 3.

*Feature Fusion*: Three layers, including the average pooling layer of inception v3, the Global average pooling layer GoogleNet, and the FC layer of DenseNet201, are used to extract the features. The approach of destination transfer learning is used to train these models. Each chosen layer’s feature vector has a size of $FV_{1}$ × 2048, $FV_{2}$ × 1000, and $FV_{3}$ × 1000. A mode value-based strategy, covered in more detail in the following section, unites these qualities.

### 3.3. Tampering Attack on Secure CNN

The main goal is to stop tampering attempts against a trained model so that the effectiveness and outcomes are not jeopardized. This article suggests a tempering approach that attempts to temper the learned model at various stages to evaluate the capabilities of the proposed Secure CNN. The combination of BC technology with CNNs is justified by the suggested assault. The proposed tampering attack’s pseudo-code is presented in Algorithm 1.

**Algorithm 1:** Model Parameter Tampering**Input:** CNNmodel, parameters**Output:** SecureModel _t_1: j ← 02: layers ← CNNmodel. layers3: attack_type_ ← [0, 1, 2]4: if (attack_type_ == 0)Classes ← findfullyConnectedLayer (CNNmodel. layers)Classes _new_ ← interchange (Classes)findSoftmaxLayer (CNNmodel. layers) = Classes_new_End if5: else if (attack_type_ == 1)attack_name_ ← averagePerform Attack_1_Layers_output_ = FindDenseLayers(CNNmodel)Layer_shape_ ← Layers_Output_.shapenoise ← GaussianNoise(ρ, Layer_shape_)w ← weights (Layers_output_)w_new_ ← w + noiseweights (Layers_output_) = w_new_End if6: else if (attack_type_ == 2)attack_name_ ← severePerform Attack_2_for j ← 0 to size (Layers)Layershape ← Layers_i_.shapenoise ← GaussianNoise(ρ, Layer_shape_)w ← weights (Layers_i_)w_new_ ← w + noiseweights (Layers_i_) = w_new_end for end if7: Securemodel_t_ = trainMod(CNNmodel)

### 3.4. Features Concatenation and Optimization

Three feature maps have been utilized namely, Inception V3 features, GoogleNet features, and Densenet201 features, denoted by $\varphi_{1}\left(FV_{1}\right)$, $\varphi_{2}\left(FV_{2}\right)$, and $\varphi_{3}\left(FV_{3}\right)$. Where $FV_{1}$, $FV_{2}$, and $FV_{3}$ constitute the dimensions of the derived features. The dimensions of the feature vectors are mentioned in Figure 4. We first serially combined all the features in one vector, as follows:(2)$$\varphi_{li}={\left(\begin{array}{c} \varphi_{1}\left(FV_{1}\right)\\ \varphi_{2}\left(FV_{2}\right)\\ \varphi_{3}\left(FV_{3}\right) \end{array}\right)}_{N\times Li}\in S$$
where $\varphi_{li}$ is the size of the final feature serially fused vector and Li stands for the number of features in each feature vector. Later, we group all the features according to their highest values, and to do this, the mode value is calculated. The features are arranged in the highest order according to the mode value. Later, the fitness function Entropy Controlled Naive Bayes for the GA was used. The GA Algorithm 2 is provided below:

**Algorithm 2:** G-A based Feature SelectionInput: 4048Output: 2988Class Label: yLaplace smoothing constant: λMaximum entropy threshold: τ1: Initialized Parameters
Initial_Population size (P) = 100Maximum_Iterations = 500Rate of Mutation = 0.001Rate of Crossover Rate = 0.8B = 6
2: Fitness function calculationCalculate the class probabilities p(y) for each class yCalculate the entropy H(y) of the class distribution using the following formula:


H(y)=−∑p(y)log(p(y))



$H_{i}=-\sum p(F_{i})\log(p(F_{i}))$


 (Where $p(F_{i})$ is the proportion of samples extracted from a population) If $H_{i}$ > τ, set the fitness score to 0 If $H_{i}$ ≦ τ set the fitness score to the initial population3: Uniform Cross-Over Implementation4: Features extraction using Roulette Wheel5: Mutation_Implementation 6: Populations_Merger7: Population_Sorting8: Robust_Chromosomes

## 4. Experimental Results

The proposed secure CNN has been employed to the publicly available brain tumor classification MRI scan dataset available on Kaggle. The dataset comprises four classes: no tumor, meningioma tumor, pituitary tumor, and glioma tumor. The number of images per class varies between 395 and 827 images per class. The sample images from the brain tumor dataset have been presented in Figure 5. The dataset is divided into two portions: training and testing. The specified CNN model was trained on an Nvidia GeForce GTX 1080 with 6.1 computing units, seven multiprocessors, and a clock speed of 1607–1733 MHz. The dataset is split into two portions using the conventional 70–30 split method: training and testing. MATLAB 2022b is used to train and test the CNN model. The Stochastic Gradient Descent with Momentum (SGDM) algorithm is the training method for 48 minibatch sizes. After every 20 epochs, the learning rate is reduced by 10 from its initial value of 0.01 to 0.001. The maximum epochs are set at 450, and the momentum is set at 0.4. As Cross-Entropy [22] has demonstrated an adequate performance for many multiclass problems, it is employed as an acceptable loss function. Different output layers are chosen for the CNN models to extract the features.

### 4.1. Brain Tumor Prediction Results Using Secure CNN Feature Fusion

Table 1 presents the results of the different classifiers (F-KNN, C-KNN, C-SVM, Q-SVM, W-KNN, C-KNN, LD) on a Brain MRI dataset with or without blockchain implementation. The metrics used to evaluate the classifiers’ performance are accuracy (Acc), precision (Pre), recall (Rec), F1 score, and time to train (TT) and predict (PT). In general, the implementation of blockchain generally leads to an improvement in the classifiers’ performance, as evidenced by the higher accuracy, precision, recall, and F1 score values. The C-SVM classifier shows the best performance, with an accuracy of 81.84% and an F1 score of 82.77% when blockchain is implemented, having less computational time. Computational time can be comprised of the security and privacy of the prediction model. The highest accuracy was achieved on the LD classifier, with an accuracy of 86.66%, a precision of 87.99%, and a recall of 85.14% with an f1 score of 85.83%.

### 4.2. Brain Tumor Prediction Results Using Secure CNN Feature Optimization

Table 2 presents the predictions outcomes of the proposed secure CNN-based model for brain tumor. The derived features from the secure CNN are fused and optimized using a genetic algorithm. The performance of the classifiers is measured using the accuracy (Acc), precision (Pre), recall (Rec), F1 score, and time taken to train (TT) and predict (PT) the model. The LD classifier, which does not use blockchain data, achieves the best accuracy (85.21%), followed by C-KNN (79.77%). C-KNN comes in second with 78.93%, and the LD classifier has the best precision with 84.96%. With 84.74%, the LD classifier achieves the highest recall, followed by C-KNN with 78.05%. W-KNN comes in second with 78.95%, while the LD classifier earns the best F1 score at 84.98%. Regarding the classifiers that use blockchain data, the LD classifier has the greatest accuracy (99.75%), followed by W-KNN (83.92%). The W-KNN and LD classifiers both achieve excellent levels of precision, with 98.97% and 81.83%, respectively. With 97.94%, the LD classifier achieves the highest recall, followed by W-KNN with 83.74%. W-KNN has the second highest F1 score (83.64%), while the LD classifier has the highest at 97.73%. Overall, the LD classifier performs the best in terms of the accuracy, recall, precision, and F1 score for both datasets, with and without blockchain information. However, it is important to note that the time taken by the classifiers to train and predict the model is also an important factor to consider in the selection of a classifier.

The learned classifiers were altered during the trials using various tempering attacks. These assaults were carried out at varying degrees of intensity. Mild, moderate, and severe assaults were classified according to their severity. Only the output classes were affected by the minor attack, whereas, in the typical attack, the output layer weights and classes were moderated. The weights of all the layers, strides, filter sizes, output size of the output layer, and output classes are altered in the severe attack. In Table 3, the outcomes of the tumor prediction using the extracted features with and without blockchain integration by applying the LD classifier are shown.

In Figure 6, the proposed approach has been individually compared with deep learning models. Each model, Inception v3, GoogleNet, and Densnet201, were fine-tuned for the tumor prediction. The maximum accuracy achieved by Inception v3 was 83.45% and GoogleNet achieved 84.27%; when all three network features were fused, the accuracy increased to 86.66%. The proposed secure CNN-based model with feature optimization achieved the highest accuracy of 99.75%. The highest accuracy shows the robustness of the proposed approach.

In Table 4, a comparison with the current methods is shown. The proposed strategy performs better than the existing strategies, as demonstrated in the table. The proposed method shows a competitive performance compared to the state-of-the-art (SOTA) methods. The proposed Secure CNN-based model utilizes blockchain to secure the CNN model layers’ inputs and outputs efficiently. The inclusion of blockchain layers in the CNN architecture increases the network complexity, but under the fine-tuning of the hyperparameters, the implanted approach proved its robustness in prediction and security.

## 5. Discussion

The CNN model for predicting brain tumors is secured in this work using an established blockchain technique. Three deep models are used, secured by a blockchain architecture, and the features are extracted. The serial mode value is used to combine the features. Later, we employ the fitness method, known as entropy-controlled naive bayes, to attempt to improve the GA. By using this method, the features are selected, and the ideal chromosomes are obtained and provided to the machine learning classifiers for final categorization. Based on the findings, it is evident that even a minor attack reduced the suggested model’s accuracy by 25.20%, and the results remained almost unchanged when the attack was made on a network using a blockchain. Similarly, the categorization accuracy was reduced by 37.90% and 59.7.98%, respectively, for average and severe assaults. These results demonstrate the validity of the suggested secure models and their resistance to tempering attempts.

## 6. Conclusions

In this work, a blockchain-based CNN model has been presented for the prediction of brain tumors using MRI scans. The proposed secure CNN-based prediction method for brain tumors has been secured by adding blockchain layers into the CNN models. The secure CNNs are later utilized for extracting the features from the brain MRI scans. The extracted feature is serially fused and optimized using GA. The optimized features map is used for the prediction of the different types of brain tumors. The highest prediction accuracy was achieved on the LD classifier, with an accuracy of 99.75%, precision 97.94%, and recall of 98.73%. The results shows that the blockchain-based Secure CNN proved robust under the different types of attacks and the recognition performance remains consistent and accurate. The feature optimization also increases the recognition performance of the proposed model by discarding the irrelevant features presented in the fused feature map. In the future, Secure CNN could be strengthened using several hashing methods and complex LLB-CNN integration.

## Figures and Tables

**Figure 1 diagnostics-13-01229-f001:**
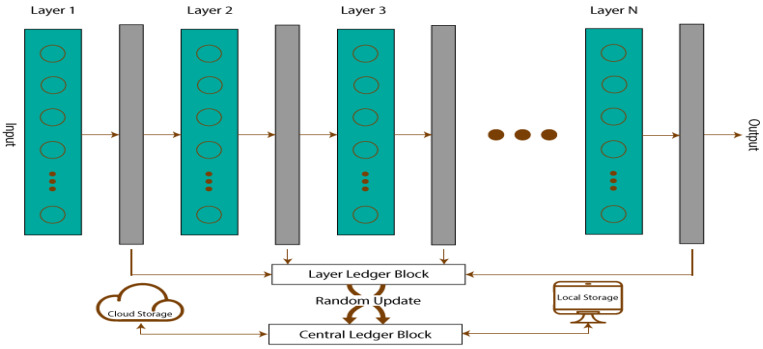
Structure of Secure CNN.

**Figure 2 diagnostics-13-01229-f002:**
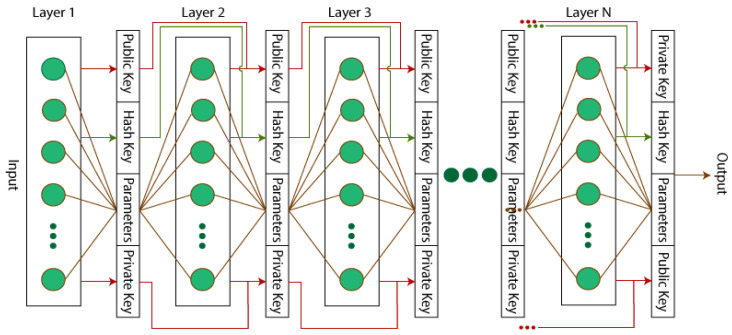
Organization of layer ledger block.

**Figure 3 diagnostics-13-01229-f003:**
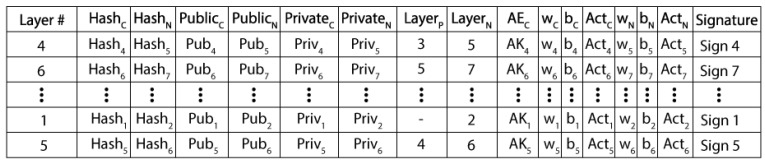
Organization of central layer ledger block.

**Figure 4 diagnostics-13-01229-f004:**
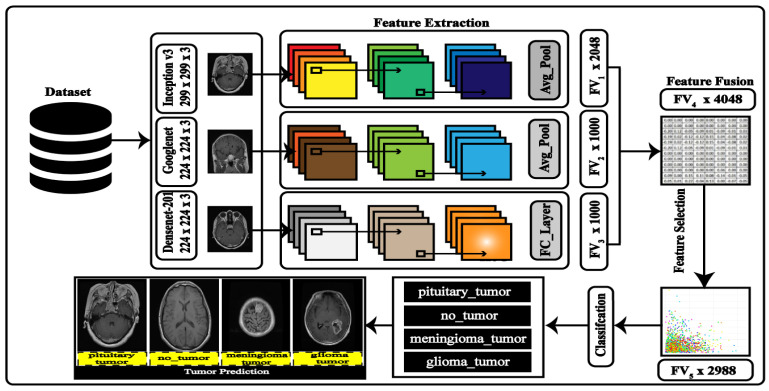
Proposed CNN-based model for brain tumor prediction.

**Figure 5 diagnostics-13-01229-f005:**
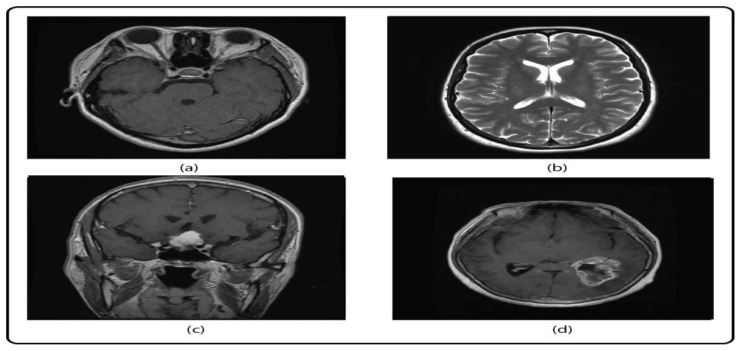
Sample brain tumors MRI scans (**a**) Pituitary (**b**) No_tumor, (**c**) Meningioma (**d**) glioma.

**Figure 6 diagnostics-13-01229-f006:**
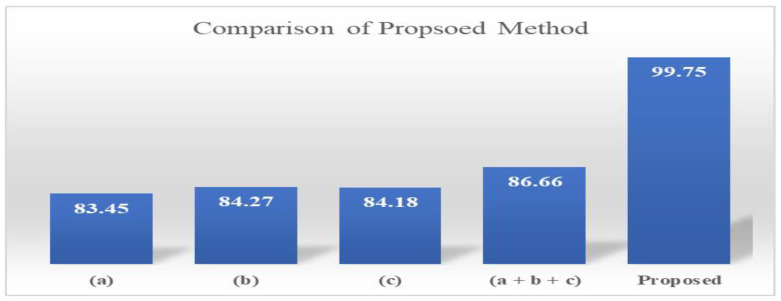
Comparison of proposed method with deep learning models (**a**) Inception v3 (**b**) GoogleNet (**c**) Densenet201 (**a** + **b** + **c**) Fused feature map.

**Table 1 diagnostics-13-01229-t001:** Brain Tumor prediction results using Secure CNN feature fusion.

Classifier	Blockchain		Acc (%)	Pre (%)	Rec (%)	F1 (%)	TT (s)	PT (s)
	No	Yes						
F-KNN	✓		41.39	41.75	40.95	42.94	247	1.54
		✓	73.64	71.66	72.54	72.04	253	1.29
C-KNN	✓		33.16	33.13	32.22	32.56	249	0.77
		✓	67.47	67.47	66.83	67.89	216	1.09
C-SVM	✓		46.81	43.19	48.64	46.47	343	2.52
		✓	81.84	80.16	81.85	82.77	378	2.08
Q-SVM	✓		49.01	50.99	47.95	48.45	249	1.41
		✓	79.91	80.09	81.91	80.73	292	1.89
W-KNN	✓		33.16	32.84	32.22	32.56	245	0.88
		✓	72.27	71.88	73.45	73.34	203	0.94
C-KNN	✓		52.87	53.13	51.05	54.53	147	0.71
		✓	82.57	83.43	81.38	83.57	133	0.54
LD	✓		57.21	57.69	55.84	57.18	193	0.35
		✓	86.66	87.99	85.14	85.83	172	0.41

**Table 2 diagnostics-13-01229-t002:** Brain Tumor prediction results using optimized Secure CNN features.

Classifier	Blockchain		Acc (%)	Pre (%)	Rec (%)	F1 (%)	TT (s)	PT (s)
	No	Yes						
F-KNN	✓		54.34	53.77	51.78	51.94	228	1.29
		✓	83.84	81.86	82.44	82.58	238	1.05
C-KNN	✓		53.16	53.82	52.22	52.56	229	0.66
		✓	77.47	77.56	76.93	77.19	189	1.09
C-SVM	✓		46.81	43.19	48.64	46.47	343	1.82
		✓	84.84	84.78	83.95	84.77	359	1.05
Q-SVM	✓		79.86	78.99	77.98	79.45	219	1.21
		✓	89.91	88.09	80.91	88.73	282	1.39
W-KNN	✓		73.06	72.64	72.78	72.99	245	0.76
		✓	83.92	81.83	83.74	83.64	203	0.94
C-KNN	✓		79.77	78.93	78.05	78.95	138	0.75
		✓	82.57	83.43	81.38	83.57	125	0.48
**LD**	✓		85.21	84.96	84.74	84.98	178	0.33
		✓	**99.75**	**98.97**	**97.94**	**98.73**	**135**	**0.31**

Bold represent the outcomes of the proposed model highest values.

**Table 3 diagnostics-13-01229-t003:** Performance of the proposed method under different types of attacks.

Attack	Blockchain		Accuracy (%)
	Yes	No	
Mild		✓	74.8
	✓		96.8
Average		✓	62.1
	✓		96.9
Severe		✓	40.3
	✓		97.0

**Table 4 diagnostics-13-01229-t004:** Proposed method comparison with (SOTA).

Reference	Year	Method	Accuracy (%)
[49]	2022	Secure Cloud Dee CNN	97.87
[50]	2022	Hybrid Deep CNN	98.1
[51]	2023	SqeezNet + SVM	98.5
[52]	2023	Deep CNN + Salp Swarm Algorithm	99.1
[53]	2023	SSO-RBNN	96
**Proposed**		**Secure CNN**	**99.75**

## Data Availability

The brain MRI scans dataset is utilized in this manuscript which is publicly available dataset on Kaggle (https://www.kaggle.com/datasets/sartajbhuvaji/brain-tumor-classification-mri). accessed on (2 February 2023).

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
