# Peer review of "Blockchain-Based Deep CNN for Brain Tumor Prediction Using MRI Scans"

_diagnostics, 2023, doi:10.3390/diagnostics13071229_

Round 1

Reviewer 1 Report

In this manuscript, the authors used a block chain-based convolutional neural network model for prediction of brain tumors by magnetic resonance imaging (MRI) scans. This work is interesting and will attract attentions of experts from medical diagnosis and treatment. Thus, I recommend the acceptance of this study after addressing the following minor issues:

1.     What are the possible interference factors for the diagnosis process? The author can give a discussion about this.

2.     Abbreviation should be used with the full name at the first time.

3.     Conclusion part should be added in this manuscript.

4.     In Figure 5, the author should label every picture with letters such as (a)(b)(c)(d) and show the detailed information in the captions.

5.     There are some language errors that should be carefully checked and revised.

Reviewer 2 Report

The authors propose a deep CNN for brain tumor prediction enhanced via blockchain technology to make  secure the output of the prediction.

Overall the proposed methodology is interesting and effective, but some important mathematical details, in my opinion, are not well explained. Here follows my comments:

-Section 3.1: Expression (1), please, specify what is r.

-Section 3.1: starting from line 240 till 245, please, be rigorous in explaining what are the needed computations. What is the symbol in front of the parenthesis at line 241? What does "Mi" denote? What does "Pubi" denote? 

- Section 3.2.1 : in the description of the trained networks appear the symbols "FV_1", "FV_2" and "FV_3" which the authors explain in section 3.3. I would suggest to write the meaning already in section 3.2.1 as it is not clear by simply looking at Figure 4.

-Section 3.2, Algorithm 1: Please, write what is r appearing as input variable inside the function "Gaussian noise" and how much it is. 

-Section 3.3, expression 2: The authors introduce expression (2) $\varphi_{li}$ as a vector, then, in the row below expression (2), the authors say that $\varphi_{li}$ is the size of the final fused vector. Hence, it seems that $\varphi_{li}$ is simply a number, which is obviously not the case. So, please, correct this statement. Also correct the notation: \varphi_{li}--> \varphi_{Li} and since FV1, FV2, FV3 are never used before, in their place the notation "FV_1", "FV_2" and "FV_3" should be used. 

-Section 3.3, Algorithm 2: explain the meaning of the used variables, i.e., q, Q, S_1, S_2, G. Also, in the fitness function definition, there is a summation over the index L but none of the variable inside the summation depends on L, hence please correct it and be rigorous otherwise the proposed algorithm is not reproducible. 

- Section 4.2: please specify that the results displayed on Table 3 are obtained by using the LD classifier (at least this is my perception). 

Finally there are some misspelled words and sentences that the authors should correct:

-Line 36 edema, Oedema. Please choose one style, either US or British English and be consistent with that.

-Line 65 "a modern machine (ML)--> a modern machine learning (ML)

-Line 76 "At each tier"--> at each iteration

-Line 262: After the full stop, "Figure 4" is missing in the text.

-Line 278: Machine Learning (ML) is already introduced before, hence, simply use ML. 

-Line 348: the acronym C-KNN is repeated. 

- Then throughout the manuscript the words "tamper", "tampered" or "tampering" are misspelled several times as "tEmper", "tEmpered" or "tEmpering" . Hence, please correct them. 
